# Characterization of Breast Tumors from MR Images Using Radiomics and Machine Learning Approaches

**DOI:** 10.3390/jpm13071062

**Published:** 2023-06-28

**Authors:** Khuram Faraz, Grégoire Dauce, Amine Bouhamama, Benjamin Leporq, Hajime Sasaki, Yoshitaka Bito, Olivier Beuf, Frank Pilleul

**Affiliations:** 1Univ Lyon, INSA-Lyon, Université Claude Bernard Lyon 1, CNRS, Inserm, CREATIS UMR 5220, U1294, 69621 Lyon, France; 2FUJIFILM Healthcare France S.A.S., 69800 Saint-Priest, France; 3Department of Radiology, Centre Léon Bérard, 69008 Lyon, France; 4FUJIFILM Healthcare Corporation, Tokyo 107-0052, Japan

**Keywords:** radiomic analysis, tumor characterization, IHC markers, automatic classification, multi-contrast MRI

## Abstract

Determining histological subtypes, such as invasive ductal and invasive lobular carcinomas (IDCs and ILCs) and immunohistochemical markers, such as estrogen response (ER), progesterone response (PR), and the HER2 protein status is important in planning breast cancer treatment. MRI-based radiomic analysis is emerging as a non-invasive substitute for biopsy to determine these signatures. We explore the effectiveness of radiomics-based and CNN (convolutional neural network)-based classification models to this end. T1-weighted dynamic contrast-enhanced, contrast-subtracted T1, and T2-weighted MR images of 429 breast cancer tumors from 323 patients are used. Various combinations of input data and classification schemes are applied for ER^+^ vs. ER^−^, PR^+^ vs. PR^−^, HER2^+^ vs. HER2^−^, and IDC vs. ILC classification tasks. The best results were obtained for the ER^+^ vs. ER^−^ and IDC vs. ILC classification tasks, with their respective AUCs reaching 0.78 and 0.73 on test data. The results with multi-contrast input data were generally better than the mono-contrast alone. The radiomics and CNN-based approaches generally exhibited comparable results. ER and IDC/ILC classification results were promising. PR and HER2 classifications need further investigation through a larger dataset. Better results by using multi-contrast data might indicate that multi-parametric quantitative MRI could be used to achieve more reliable classifiers.

## 1. Introduction

The identification of various immunohistochemical (IHC) markers, like the status of estrogen (ER) or progesterone (PR) hormones, or the responsiveness of the tumors to the HER2 protein, is helpful in determining the course of treatment in breast cancer patients. The overexpression of ER and PR receptors is observed in numerous breast cancer cases, and this characteristic has been helpful in developing targeted treatments [1]. HER2 (human epidermal growth factor) expression also plays an important role in breast cancer prognosis, and its overexpression is associated with malignancy [2]. HER2 has been used to design strategies for breast cancer treatment [3]. A combination of the statuses of various IHCs is used to describe different molecular subtypes, of which luminal A (ER^+^ and/or PR^+^, HER2^−^, low proliferation marker Ki-67) and luminal B (ER^−^ and/or PR^+^, HER2^+^ or HER2^−^ with high Ki-67) are frequently used for characterizing breast tumors [4]. Another widely used tumor characterization is triple-negative (TN), where ER, PR, and HER2 statuses are all negative. Histologically, breast tumors may be classified as invasive ductal carcinoma (IDC), invasive lobular carcinoma (ILC), or a combination of the two. The histological type influences the prognosis in conjunction with the molecular subtypes [5,6]. Traditionally, genetic approaches have been used to determine the molecular subtypes in breast cancer tumors. This involves the analysis of a sample obtained through a biopsy. Throughout the past decade, there has been a continuous and ongoing exploration of the relationship between radiomic features and the molecular subtypes of breast cancer [7,8,9,10,11].

A radiomic analysis involves extracting a number of features describing the shape, size, or texture of a region selected in an image. Several existing studies have explored the relationship between radiomics features and various IHC markers, or, on a more basic diagnostic level, to distinguish benign and malignant tumors. Radiomics, combined with machine learning, is used in [12] to distinguish benign satellite tumors from malignant ones. The markers extracted from DCE (dynamic contrast-enhanced) MR images were used in [13] to differentiate between invasive and noninvasive breast lesions. This study was also extended to the classification of the malignancy level of breast tumors [14]. The DCE MR images were used in [15] for the benign vs. malignant classification of breast tumors. In another study, Chou et al. target the ductal carcinoma cases for predicting their receptor statuses through the features in DCE MR images [16]. Tan et al. [17] used radiomics to predict auxiliary lymph node metastasis. Radiomics analysis has also been used to complement the traditional genetic-based analysis [18]. It should be noted that the use of radiomics-based analysis is not exclusive to the MR image modality. For instance, radiomics analysis on echographic images has also been used to characterize breast tumors [19,20]. Furthermore, instead of extracting the radiomics, the deep learning approaches have also been directly applied to the image data [21].

There has been an upward trend in the radiomics-based prediction of the molecular subtypes [11,22] and continues to be a subject of interest [23,24,25,26]. Multi-contrast MR images were used in [27] for a radiomics-based prediction of HER2 status and TN cases. Radiomics is used for TN prediction on mono-contrast images in [28]. Xie et al. [29] also used radiomics, but on multi-parametric data for TN prediction. In some studies, the HER2 status is reported to be difficult to predict through radiomics analysis on the regular contrast images, but a potential for improvement for this classification by using multi-parametric data is shown [30,31]. There are fewer studies that use radiomics to classify individual IHC markers, as opposed to a direct characterization of molecular subtypes defined by them. Among such studies, ER and PR classification, along with HER2 classification on MR images is performed by Li et al. [31]. ER, PR, and HER2 prediction in the context of studying breast cancer brain metastasis is the subject of a study by Luo et al. [32]. The classification of ER, PR, and HER2 is conducted on radiomics and genomics data by Yoon et al. [33]. In a recent work by Zhong et al. [34], radiomic features from parametric maps in MRI were used to predict the ER and PR statuses. While a vast majority of the tumors are of type IDC, ILC remains understudied [35,36] and there appears to be a limited existing literature on the radiomics-based differentiation of the two types. A textural analysis of the breast tumors was used by Holli et al. [37] to differentiate IDC and ILC. Both texture and entropy features were individually used by Waugh et al. for the same task [38].

The aim of this study is to explore the predictability of various molecular signatures as well as the histological subtypes IDC and ILC in breast tumors via both a mono- and multi-contrast radiomics analysis and, in a limited case, compare it with a CNN (convolutional neural network)-based approach. Although the HER2 status is frequently explored in the literature, we found fewer studies that used radiomics for predicting the ER and PR responses individually, rather than as a part of luminal A, luminal B or TN molecular subtypes. In the existing literature on radiomics-based analysis, the histological subtypes have not been as much investigated as the molecular subtypes or the IHC markers. In our study, the IDC/ILC classification is performed on the same cohort of patients as used for the ER, PR, and HER2 classification. A partial contribution of our study is to group these tasks, which can individually be found in the existing literature.

## 2. Materials and Methods

### 2.1. Image Acquisition Protocol

This study included patients who had a diagnosis of breast cancer confirmed both by a biopsy and via a breast MRI (median age: 52 years; age range: 24–86 years; female patients only). Furthermore, the patients were included in the study only if they did not present a history of thoracic surgery, chemotherapy, or radiation therapy. Only the baseline MRI at diagnosis was included before any chemotherapy or surgery. Patients were included after their biopsy (range: 2–28 days). The MRI protocol was sequentially composed of T1-weighted imaging (T1), Dixon T2-weighted imaging (in-phase, out-of-phase, water and fat images) or T2 SPAIR (T2), and 3D gradient-echo dynamic contrast-enhanced (DCE) images with fat suppression, acquired at 3 and 5 min after the intravenous injection of 0.1 mmol/kg gadolinium-based contrast agent. Furthermore, a subtraction image (Sub) was achieved through the subtraction of the T1 image from the DCE image. The acquisitions were performed using three different MR systems at two different static magnetic fields (Philips Ingenia 1.5 T, SIEMENS MAGNETOM Vida 3.0 T, and SIEMENS MAGNETOM Aera 1.5 T). The data acquisition period is from early 2017 to mid-2020. Histological subtypes, HER2 expression, ER expression, and PR expression were retrieved from the patients’ database. Patient data were collected after institutional review board approval. Patients who did not consent to the use of their clinical data for an academic study were excluded, according to national and European laws.

### 2.2. Segmentation of the Lesions

The lesion/tumor region in the images needs to be separated before performing a radiomics analysis. Ideally, this should be fully automated to allow for an efficient treatment of a large dataset, which is crucial for machine learning-based analysis. However, at this stage, we used segmentations overseen by an experimented radiologist, with 9 years of experience in oncology imaging, so that the impact of any segmentation errors on the radiomics analysis is minimized. A semiautomatic tool available in ITK-SNAP [39] was used for the guided segmentation of the tumors. The lesions were delineated on DCE images, in which the pixels corresponding to a tumor tend to have a higher intensity than the neighboring pixels. The guided steps allowed for an approximate segmentation of the tumor within an ROI box drawn around the lesion. An experimented radiologist subsequently made slice-by-slice manual correction of the segmentation mask.

### 2.3. Data Preparation and Experimental Setup

Some small artifacts in the masks (like thin gaps or uneven boundaries) introduced due to manual delineation of the tumor were smoothed out through simple mathematical morphology operations (sequential opening and closing using a 3 × 3 × 3 pixels cubic kernel). Among the T2-weighted acquisitions, fat-suppressed (with Dixon fat–water decomposition or spectral attenuated inversion recovery (SPAIR)) images were selected for radiomics analysis. All three image types (DCE, Sub, and T2) were extrapolated to the same dimensions using ITK libraries [40]. The same segmentation mask, with necessary extrapolation, was applied to the three image types. We excluded the tumors having very small sizes (smaller than 5 mm) because they are potentially inadequate for a radiomics analysis and are subject to particular consideration [41]. Indeed, the dimensions of some convolution kernels come very close to the size of small tumors. The tumors with roots spread in the entire gland were also excluded due to their complex bifurcations and very irregular shape, which pose difficulty in precise segmentation.

For experimental purposes, we divided our data into five sets (folds) by random selection of patients. Initially, we established the feasibility of the radiomics-based classification of molecular subtypes by training different machine learning models on four of the five folds to maximize the classification precision on the fifth one. In a second setup, we reserved one fold as a pseudo-external validation dataset (referred to as a test dataset in the article). After training the model on three folds to obtain optimum results on a fourth layer, the model was applied to the test dataset. We also tried to determine the impact of the MRI contrast on the classification task. Therefore, the experiments were performed either on the individual types of images (mono-contrast) or on a combination of types (multi-contrast). In the case of mono-contrast, the data from only one type of image (DCE, T2, or Sub) were used, whereas the data from two or three types of images were combined in multi-contrast experiments.

We used the setup with a test dataset in two other types of experiments. One set of these additional experiments was to explore the performance of the CNN-based approach, for which we compared the performance on the mono-contrast data with the radiomics approach. The second set of the exploratory experiments applied the radiomics-based approach on a dataset having a stricter exclusion criterion regarding the tumor sizes to have a more homogeneous data in this regard. For this, we retained only the tumors with a size within 50% of the median size of the tumors in the previous dataset. We used the average of the length, width, and height of a tumor in the orientation defined by the acquisition setup as a representative of its size. Since the tumors may be quite irregular in shape, we used this indicator of size, rather than the precise 3D volume, for a basic statistical analysis of the homogeneity of tumor sizes.

### 2.4. Radiomics-Based Classification

Figure 1 illustrates a radiomics-based pipeline for classifying the molecular subtypes. PACS is the database that contains the complete patient scans. The images, along with the histology results, obtained from this database are anonymized and annotated. Once the tumors have been segmented, a radiomics analysis is performed over the region corresponding to the segmentation mask. The details of the extracted features can be found in [42]. Figure 2 summarizes the categories of the 342 radiomics features extracted. Since the initial radiomics set contains several hundred features, a dimension reduction algorithm is applied to select the most pertinent and uncorrelated radiomic features. A classification model is subsequently trained on the reduced number of radiomic features. The approaches used in the key steps of this pipeline are presented in detail below.

The radiomic features were extracted from the zones of the tumors corresponding to the manually segmented masks. From each image, 6 contrast images were derived by using different intensity levels (8, 16, 24, 32, 48, and 64). This allows for taking into account the intensity variation in the original image due to variations in acquisition conditions. The radiomic features extracted from these six contrast images were then averaged. A total of 342 radiomic features describing the tumor’s size, shape, or texture, were extracted. All of these radiomic features are not necessarily relevant, and using all of them might result in an ineffective model. Hence, a dimension reduction was needed. Two kinds of dimension reduction algorithms were tested: (i) ReliefF [43]—selection of the least correlated features and (ii) LASSO (least absolute shrinkage and selection operator)—regression-based.

For ReliefF, 32 most relevant features were selected, whereas a relaxed reduction weight for LASSO was selected over a hundred features. The reduced features were selected by using only the train and validation datasets (the data were also z-score normalized with respect to the combined train and validation datasets before the dimension reduction). The list of features thus obtained was then used to select the same features for the test dataset as well.

Two types of classification models were used: (i) support vector machine (SVM, with a linear kernel and no regularization) for a linear classification and (ii) a fully connected neural network (FCNN) with five layers (with the number of neurons in these layers being 50, 40, 30, 20, and 10). Different combinations of dimension reduction approaches and classification models were used. For each experiment, one of the five folds was retained as the test dataset. For the remaining data, 25 models for each experiment were generated by randomly dividing the train–validation dataset into 25% validation and 75% train dataset (considering that one-fifth of the data was retained for the test, this amounts to a 20/20/60 split into test/validation/train). Out of 25 models, the one with the optimum performance on the validation dataset was retained for a given test fold. The results were averaged over 5 cycles by choosing different test folds.

### 2.5. CNN-Based Classification

As a second approach to classifying tumor subtypes, we used a CNN-based classification scheme applied directly to the manually segmented tumors. This was achieved through an adaptation of faster R-CNN architecture [44] that employed a dense U-Net [45] with 3 resolution levels for feature extraction. Since a preliminary comparison of the 3D and 2D classifications did not indicate any notable differences in the performance, we opted for a 2D model due to significantly less computation time and memory overload. We first trained the region proposal networks (RPN) of faster R-CNN for lesion detection tasks on patches of 384 × 384 pixel size. The patches were randomly sampled from the larger MR images with the constraint that 90% of them should contain a lesion. Squares of sizes 5, 10, 25, 50, and 100 pixels were used as anchors for the RPN. We used distance-IoU loss [46] for bounding-box regression and focal loss for classification. The detection results achieved an average recall of 90% across 5 cross-validation folds at the cost of an average of 10 false positives per patient. We used these pretrained RPN to train a full faster-RCNN architecture on the tumor subtype classification task. After the feature maps were computed, we extracted the features corresponding to a given region of interest with the ROI align technique [47]. Since reducing the number of parameters helps to prevent overfitting, we applied adaptive average pooling to reduce the multi-dimension vector to a 1D vector. We mapped a single linear layer on this vector that had the number of classes to be predicted as output. We used the same 5-fold data split scheme as used for the radiomics approach for training and testing the CNN-based classification model. All experiments were performed using Adam optimizer with default parameters [48].

## 3. Results

After presenting the dataset used in the experiments, the AUC (short for AUROC, the area-under-the-receiver–operator curve) results for different experimental setups are given.

### 3.1. Dataset

Although a database of 534 tumors (from 367 patients) was available, the exclusion of 78 tumors due to some factors, such as incomplete histological data, limited the number of tumors available in this study. Moreover, we excluded twenty-five tumors due to their small size (smaller than 5 mm) and two tumors due to their very large size (spread in the entire gland). For comparison purposes, only the tumors for which all three types of acquisitions (DCE, Sub, and T2) were available were retained. These exclusion and selection criteria left us with 429 tumors from 323 patients.

Figure 3a summarizes the number of tumors for both categories of different classification tasks addressed in this study. Not only was there a large imbalance in classes for all tasks, but the tumors also displayed wide heterogeneity in other aspects as well, as Figure 3b illustrates for tumor sizes. The tumor size distribution was 24.6 ± 16.6 mm (mean ± SD), with a median value of 20.3 mm, around which most of the tumors are located. Please note that each of the ER, PR, HER2, and IDC/ILC statuses is not available for all the tumors. Hence, a disparity in the number of total number of tumors and the number of samples available for a given classification task is shown in Figure 3a. Figure 3b, on the other hand, is a histogram of the tumor size distribution of all 429 tumors.

### 3.2. Classification Results

Table 1 presents the results for different approaches without setting apart a test dataset. The results are obtained over random 80/20 training/validation splits. The results shown are the ones with the best AUC of the 25 models for the radiomics-based approach. For the experiments involving results given in Table 2, the dataset is divided into five patient-wise folds, as described in Section 2.4. This table displays results both for the radiomics-based and R-CNN-based approaches for various input images. To consider the impact of the tumor size heterogeneity, Table 3 shows the results of classification tasks performed using only the tumors of mean size 20 ± 10 mm. Table 4 presents category-wise numbers of radiomic features selected by the ReleifF method for a set of experiments involving T2 images.

## 4. Discussion

### 4.1. Findings

The proportion of positive and negative responses of different molecular signatures in our dataset does not diverge significantly from their occurrence reported in the general population [49,50,51]. Nevertheless, there is a noticeable unbalance between the two classes for all the IHCs and histological types. This can cause a strong bias toward the larger class during the training of a neural network model for their prediction, and, consequently, render it largely imprecise. This observation makes the AUC a more useful metric for appreciating the classification results on these data. The results in Table 1 present an insight into the potential of the radiomics-based classification. The best results on the validation data were obtained for ER^+^ vs. ER^−^ and IDC vs. ILC classification tasks, with the AUC surpassing 0.8 for several classification models. Furthermore, the results on the multi-contrast data tended to be better than on the mono-contrast alone. The results obtained using LASSO-based reduction and FCNN-based classification schemes were slightly better than those obtained by using ReliefF and SVM. For the classification tasks involving HER2 and PR, the results were significantly poorer, with AUCs around 0.6 in the majority of the experiments.

At present, we do not have access to data for external validation. Hence, for a rigorous evaluation (at the expense of reducing our training dataset) we set aside a part of our data as pseudo-external data, which we will herein refer to as the test dataset. For this setup, we also compare the performance with an R-CNN-based scheme, applied directly to the image data. The relative performance of various classification tasks in this setup (see Table 2) was in agreement with the previously discussed validation results. The experiments were most successful for ER and IDC vs. ILC tumor classification tasks, with AUC often approaching or surpassing 0.7 in both cases. The results were most unsatisfactory for the prediction of HER2 and PR statuses. In the case of multi-contrast experiments, an improvement in the ER and IDC/ILC (especially ER) classification was observed. The improvement with T2+Sub or DCE + T2 + Sub combinations was more pronounced, with the ER classification AUC reaching 0.78. On the other hand, the performance of HER2 and PR classification tasks did not improve with any of the multi-contrast combinations.

The performance of the R-CNN-based approaches is comparable to that of the radiomics-based one. Once again, a higher AUC of around 0.7 was reached for ER classification. For the IDC vs. ILC classification, the AUC was relatively lower than that of the radiomics-based approach. This difference might be explained by the lack of sufficient data to train deep-learning models. Training on a large dataset is needed to efficiently extract the relevant features directly from the images. The radiomics-based approach, on the other hand, uses mathematically formulated radiomic features. As for the computation times, the radiomics-based approach took from a few seconds to 1 or 2 min, depending on the classification methodology used, whereas the R-CNN approach required several hours to train a model even on the images resampled to a smaller size.

In the case of using the tumors with limited size variations (Table 3), while the general tendency for the ER and IDC/ILC classification was about the same as in the previous results, there was an improvement in the HER2 and PR classification results: for both cases, the results showed a tendency to approach or even sometimes surpass 0.6 AUC. This is in contrast to the previous results, where the AUC was not much different from that of a random classifier. This might indicate that the tumor heterogeneity influences the classifier and, consequently, an extensive training set covering vast variations in tumor sizes might be helpful in training a more reliable classifier.

Finally, we add a brief discussion on the radiomic features selected for the classification tasks. The radiomic features summarized in Table 4 correspond to only one set of experiments involving ReliefF-based dimension reduction that we conducted to obtain the results presented in this work. One of the salient observations is regarding the categories of features retained for the IHC classification tasks (involving ER, PR, and HER2) and the tumor-type classification (IDC vs. ILC). While all the classification tasks selected more texture features than the size and shape features, there was an overwhelmingly larger selection of texture features (30 out of 32) for the tumor-type classification. Furthermore, the texture features for the ER and PR classification tasks generally included several frequency-based features (Gabor filter responses), whereas the IDC vs. ILC classification favored the gray-level matrix-based texture features. The next step after the identification of the pertinent radiomic features is to analyze their repeatability. There have been some preliminary studies on both contrast images and multi-parametric MRIs [52,53] in this regard. While a detailed presentation of the features selected in our various experiments would be quite voluminous, we do note a few key observations. The features selected depended not only on the classification task, but also on the contrast types used. The repeatability of features in the tasks with a higher classification accuracy was considerably higher than the other classification tasks. For instance, a large majority of the radiomic features remained the same in the case of ER^+^ vs. ER^−^ classification task for five experiments for different folds of the training dataset. However, less than half were reproducible for HER2^+^ vs. HER2^−^ classification task.

### 4.2. Comparison with Existing Studies

Although our performance level and results of the MRI radiomics-based IHC classification task do not consistently match with those found in the existing literature, the tendency of ER status to be more easily predictable and the better classification found using multi-contrast data are in agreement with the existing literature. In a study performed by Li et al. [31] on the TCGA dataset, the AUCs for ER, PR, and HER2 classification tasks were 0.89, 0.69, and 0.65, respectively. In a more recent study, Luo et al. [32] reported promising results in a breast cancer brain metastases study, with ER, PR, and HER2 being 0.89, 0.88, and 0.87, respectively, on a cohort of 68 patients. In their comparison of radiomics-based and genomics-based classifications, Yoon et al. [33] reported the AUC values for the radiomics-based classification of ER, PR, and HER2 as 0.82, 076, and 0.72, respectively. For the ER and PR classification tasks, Zhong et al. [34] obtained their respective AUCs as 0.76 and 0.81 by using the radiomic features from the intratumoral region, and 0.73 and 0.71 for the peritumoral region. The difficulty of obtaining a better performance for HER2 classification is also reported in a previous study [30]. They worked with an extensive initial radiomic feature set of over one thousand features and obtained a better performance (AUC 0.86) using multi-contrast data. Huang et al. [54] obtained an AUC of 0.86 in leave-one-out cross-validation (LOOCV) for HER2 prediction that reached 0.86, using radiomics analysis on multi-parametric images.

As for the classification of the histological subtypes, Waugh et al. [38] obtained a 0.75 AUC using co-occurrence matrix features and a 0.63 AUC using entropy features for the classification of 92 IDC and 45 ILC tumors by using a K-nearest neighbor (kNN) classifier. Their results, whose AUC values vary between 0.62 and 0.75, are very close to ours depending on the classification method and the image types used. Holli et al. [37] differentiated 10 IDC and I0 ILC cases by using co-occurrence matrix features by using kNN and other methods. They achieved an accuracy between 80 and 100% depending on the classification method and the types of MRI used.

### 4.3. Limitations

In addition to the need to further investigate the PR and HER2 classification tasks, this study has some limitations. The dataset for all the classification tasks is very unbalanced. This is the reason we have used only the AUC and not the metrics such as precision and recall to assess our results as the last two metrics are likely to be biased by the over-represented class. It would be interesting to have obtained the results with a balanced dataset. Since the class unbalance agrees with the distribution of the studied molecular subtypes in the general population, such data would likely need to be sampled from a larger cohort of patients by discarding some data from the over-represented class. Furthermore, we have only analyzed the inhomogeneity in terms of tumor sizes only. A thorough study of the impact of various characteristics, like texture and shape, would be helpful in the selection of more pertinent radiomic features. Additionally, for a practical application, the radiomics analysis should be robust to the acquisition setup [55]. The data used in this study come from three different MR systems at two different static magnetic fields. At this stage, we have not yet developed a method to select only the radiomics features that remain independent of the acquisition setup.

## 5. Conclusions

In this work, we have shown that multi-contrast MRI-based radiomics could bring insights into the histological subtype classification (IDC vs. ILC) and ER molecular subtype classification of breast cancer. In addition, the CNN-based methods yielded results comparable to the radiomics-based classification. The diagnostic performances for the classification of PR and HER2 remain insufficient and further studies need to be performed to try to improve the performances by using other sources of radiomics such as multi-parametric quantitative MRI. The multi-parametric approach might also be helpful in investigating the repeatability of the radiomics features independent of the contrast types. Furthermore, model training on a more extensive dataset and validation on external data are highly desirable. Finally, a fully automated precise segmentation of the tumors is crucial for efficiently treating large data.

## Figures and Tables

**Figure 1 jpm-13-01062-f001:**
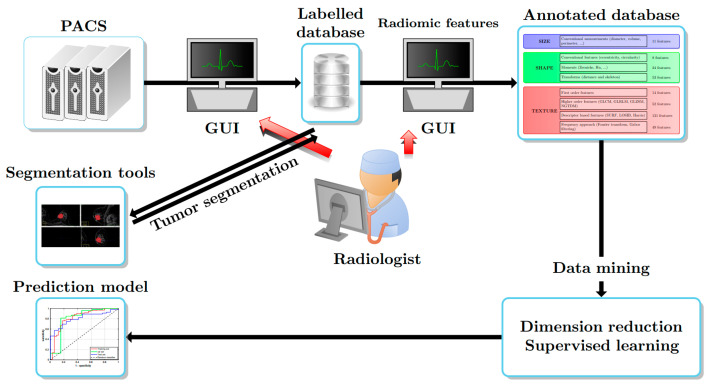
Radiomics pipeline.

**Figure 2 jpm-13-01062-f002:**
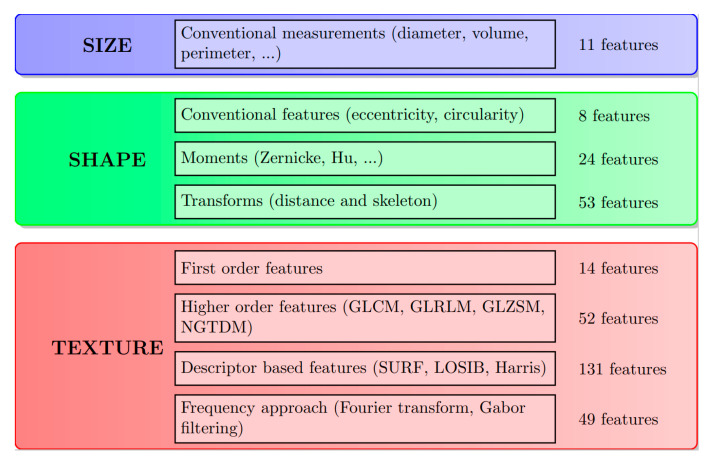
Radiomic feature set before dimension reduction.

**Figure 3 jpm-13-01062-f003:**
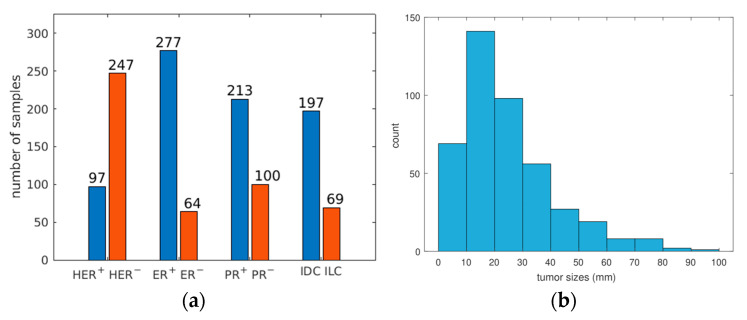
(**a**) Sample sizes of various IHC markers (**b**) Histogram of tumor sizes.

**Table 1 jpm-13-01062-t001:** AUC results for various classification models without test dataset holdout.

Classification Task	Methodology	Mono-Contrast (Val. Data AUC)			Multi-Contrast (Val. Data AUC)			
		DCE	Sub	T2	DCE + Sub	DCE + T2	Sub + T2	DCE + Sub + T2
HER2^+^ vs. HER2^−^	ReliefF + FCNN	0.61	0.66	0.69	0.65	0.65	0.61	0.57
	ReliefF + SVM	0.61	0.62	0.60	0.59	0.60	0.59	0.61
	LASSO + FCNN	0.61	0.60	0.64	0.67	0.59	0.64	0.65
	LASSO + SVM	0.63	0.70	0.63	0.61	0.59	0.65	0.61
ER^+^ vs. ER^−^	ReliefF + FCNN	0.80	0.78	0.81	0.77	0.81	0.87	0.87
	ReliefF + SVM	0.77	0.73	0.78	0.80	0.83	0.76	0.80
	LASSO + FCNN	0.85	0.87	0.85	0.90	0.88	0.91	0.93
	LASSO + SVM	0.86	0.84	0.85	0.88	0.89	0.91	0.95
PR^+^ Vs. PR^−^	ReliefF + FCNN	0.59	0.67	0.65	0.69	0.69	0.68	0.58
	ReliefF + SVM	0.63	0.61	0.64	0.58	0.67	0.64	0.66
	LASSO + FCNN	0.42	0.61	0.71	0.55	0.57	0.63	0.59
	LASSO + SVM	0.63	0.68	0.55	0.56	0.65	0.59	0.64
IDC vs. ILC	ReliefF + FCNN	0.78	0.80	0.82	0.79	0.79	0.80	0.83
	ReliefF + SVM	0.74	0.73	0.80	0.80	0.85	0.84	0.81
	LASSO + FCNN	0.79	0.74	0.87	0.80	0.85	0.83	0.80
	LASSO + SVM	0.78	0.86	0.83	0.76	0.85	0.78	0.82

**Table 2 jpm-13-01062-t002:** AUC results for various classification models on an unseen test dataset.

Classification Task	Methodology	Mono-Contrast (Test Data AUC)			Multi-Contrast (Test Data AUC)			
		DCE	Sub	T2	DCE +Sub	DCE + T2	Sub + T2	DCE + Sub + T2
HER2^+^ vs. HER2^−^	ReliefF + FCNN	0.57	0.51	0.53	0.48	0.51	0.49	0.53
	ReliefF + SVM	0.57	0.58	0.53	0.50	0.53	0.49	0.53
	LASSO + FCNN	0.53	0.53	0.50	0.49	0.52	0.53	0.56
	LASSO + SVM	0.52	0.53	0.50	0.50	0.49	0.56	0.53
	Faster R-CNN	0.51	0.53	0.45				
ER^+^ vs. ER^−^	ReliefF + FCNN	0.68	0.66	0.71	0.67	0.74	0.72	0.64
	ReliefF + SVM	0.60	0.61	0.55	0.66	0.65	0.68	0.66
	LASSO + FCNN	0.67	0.68	0.64	0.70	0.66	0.73	0.78
	LASSO + SVM	0.73	0.70	0.72	0.74	0.74	0.74	0.73
	Faster R-CNN	0.70	0.68	0.72				
PR^+^ vs. PR^−^	ReliefF + FCNN	0.52	0.53	0.49	0.57	0.50	0.58	0.55
	ReliefF + SVM	0.49	0.58	0.52	0.51	0.50	0.53	0.55
	LASSO + FCNN	0.50	0.49	0.53	0.52	0.54	0.51	0.49
	LASSO + SVM	0.51	0.49	0.52	0.48	0.47	0.55	0.51
	Faster R-CNN	0.54	0.53	0.49				
IDC vs. ILC	ReliefF + FCNN	0.64	0.67	0.73	0.67	0.67	0.70	0.68
	ReliefF + SVM	0.64	0.66	0.70	0.68	0.73	0.71	0.70
	LASSO + FCNN	0.61	0.68	0.64	0.64	0.68	0.67	0.68
	LASSO + SVM	0.62	0.65	0.69	0.69	0.63	0.64	0.69
	Faster R-CNN	0.58	0.51	0.56				

**Table 3 jpm-13-01062-t003:** AUC of test data results for various prediction models for small variation in tumor size.

Classification Task	Methodology	Mono-Contrast (Test Data AUC)			Multi-Contrast (Test Data AUC)			
		DCE	Sub	T2	DCE + Sub	DCE + T2	Sub + T2	DCE + Sub + T2
HER2^+^ vs. HER2^−^	ReliefF + FCNN	0.58	0.61	0.57	0.55	0.55	0.59	0.55
	ReliefF + SVM	0.56	0.55	0.64	0.57	0.58	0.62	0.55
	LASSO + FCNN	0.48	0.56	0.51	0.57	0.53	0.58	0.52
	LASSO + SVM	0.55	0.55	0.59	0.58	0.56	0.58	0.53
ER^+^ vs. ER^−^	ReliefF + FCNN	0.69	0.65	0.79	0.69	0.68	0.72	0.74
	ReliefF + SVM	0.73	0.70	0.8	0.64	0.77	0.75	0.73
	LASSO + FCNN	0.70	0.60	0.73	0.64	0.69	0.73	0.64
	LASSO + SVM	0.71	0.71	0.74	0.71	0.75	0.69	0.74
PR^+^ vs. PR^−^	ReliefF + FCNN	0.53	0.49	0.56	0.51	0.59	0.53	0.53
	ReliefF + SVM	0.52	0.54	0.60	0.59	0.59	0.52	0.57
	LASSO + FCNN	0.54	0.55	0.55	0.46	0.53	0.51	0.49
	LASSO + SVM	0.48	0.56	0.53	0.41	0.53	0.51	0.51
IDC vs. ILC	ReliefF + FCNN	0.68	0.62	0.70	0.64	0.65	0.67	0.74
	ReliefF + SVM	0.70	0.63	0.68	0.65	0.70	0.70	0.70
	LASSO + FCNN	0.67	0.62	0.69	0.66	0.67	0.70	0.75
	LASSO + SVM	0.69	0.64	0.69	0.70	0.69	0.65	0.71

**Table 4 jpm-13-01062-t004:** The types of radiomic features retained for various classification tasks after dimension reduction through ReliefF on data obtained from T2 images. The result is given for a single experiment chosen arbitrarily. The input data and their division into the training, validation, and test folds are the same for each classification task.

Classification Task	Number of Reduced Radiomic Features by Categories	
	Size and Shape	Texture
HER2^+^ Vs. HER2^−^	Conventional measurements and features: *n* = 10 Moments: *n* = 5	Descriptor-based features: *n* = 2; Gray level co-occurrence matrix; (GLCM) features: *n* = 15
ER^+^ vs. ER^−^	Conventional measurements and features: *n* = 3 Transforms: *n* = 1	Descriptor-based feature: *n* = 2; Gabor filter responses: *n* = 3; Other frequency-based features: *n* = 17; GLCM features: *n* = 6
PR^+^ vs. PR^−^	Conventional measurements and features: *n* = 3 Moments: *n* = 4 Transforms: *n* = 1	Descriptor-based features: *n* = 3; Gabor filter responses: *n* = 4; Other frequency-based features: *n* = 4; GLCM features: *n* = 11; Neighboring gray tone difference matrix (NGTDM) features: *n* = 1; Gray level run length matrix (GLRLM) features: *n* = 1
IDC vs. ILC	Conventional measurements and features: *n* = 2	GLCM features: *n* = 19; GLRLM features: *n* = 1; NGTDM features: *n* = 5; Gray level size zone matrix; (GLSZM) features: *n* = 5

## Data Availability

The patient data used in this study are not in open access to conform with the patient confidentiality as per the General Data Protection Regulation of the EU.

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
