# Peer review of "Characterization of Breast Tumors from MR Images Using Radiomics and Machine Learning Approaches"

_jpm, 2023, doi:10.3390/jpm13071062_

Round 1
Reviewer 1 Report
General comments:
Introduction – the authors mention that “Recently, a relationship between radiomic features and molecular subtypes of breast cancer has been reported [Grimm et al 2015].” This is not a recent work (being published in 2015); furthermore, there are several recent reports on this topic. I advise the authors to read the recent meta-analysis on radiomic differentiation of breast cancer molecular subtypes that incorporates 41 studies:
https://pubmed.ncbi.nlm.nih.gov/34624649/
Afterwards, a critical evaluation of the current work against the existing literature must be made, highlighting the novelty of the current work. This aspect should be explained in the Discussion as well.
Results: the authors mention several times the ‘median value of the mean …’ This is very confusing / erroneous. You either report the median value (which is the middle value of a dataset from smallest to greatest) or the mean value (which represents the average of a dataset). Please clarify these aspects throughout the article.
The Discussion must include a section on the limitations of this study – for instance, the large inhomogeneities found among tumour samples, as well as the reduced dataset due to the set inclusion/exclusion criteria are limitations that could impact on the overall analysis and data interpretation.
Specific comments:
Line 43 – replace ‘zone’ with ‘region’
Line 68 – correct ‘intraductal carcinoma’
Line 69 – replace ‘mon-‘ with ‘mono-‘
Line 137 – of the median or of the mean sizes?
Minor revision is required.
Author Response
Note: The line numbers referred to in the responses are from the revised manuscript version with the changes tracked. The answers are in blue.
Introduction – the authors mention that “Recently, a relationship between radiomic features and molecular subtypes of breast cancer has been reported [Grimm et al 2015].” This is not a recent work (being published in 2015); furthermore, there are several recent reports on this topic. I advise the authors to read the recent meta-analysis on radiomic differentiation of breast cancer molecular subtypes that incorporates 41 studies:
https://pubmed.ncbi.nlm.nih.gov/34624649/
Response: There is indeed an extensive literature on the differential analysis of molecular subtypes using radiomics. Although a combination of ER, PR, and HER2 statuses is used to classify breast tumors into categories such as luminal A or luminal B, we found fewer studies using radiomics to differentiate the estrogen and progesterone responses individually. In the reviewed version, we have broadened the scope of the literature review to add more recent studies and review articles (references [8-11, 22-26] on lines 53, 72). We have also added some references for the clinical context of molecular and histological subtypes (references [4-6] on lines 45, 49).
[4]. K. S. Johnson, E. F. Conant, M. S. Soo, Molecular subtypes of breast cancer: a review for breast radiologists, Journal of Breast Imaging, 3 (1) (2021) 12–24. https://doi.org/10.1093/jbi/wbaa110
[5]. T. Du, L. Zhu, K. M. Levine, N. Tasdemir, A. V. Lee, D. A. Vignali, B. V. Houten, G. C. Tseng, S. Oesterreich, Invasive lobular and ductal breast carcinoma differ in immune response, protein translation efficiency and metabolism, Scientific Reports, 8 (1) (2018). https://doi.org/10.1038/s41598-018-25357-0
[6]. Y. Xiao, D. Ma, M. Ruan, S. Zhao, X.-Y. Liu, Y.-Z. Jiang, Z.-M. Shao, Mixed invasive ductal and lobular carcinoma has distinct clinical features and predicts worse prognosis when stratified by estrogen receptor status, Scientific Reports, 7 (1) (2017) 10380. https://doi.org/10.1038/s41598-017-10789-x
[8]. S.-H. Lee, H. Park, E. S. Ko, Radiomics in breast imaging from techniques to clinical applications: a review, Korean Journal of Radiology, 21 (7) (2020) 779. https://doi.org/10.3348/kjr.2019.0855
[9]. D.-M. Ye, H.-T. Wang, T. Yu, The application of radiomics in breast MRI: a review, Technology in Cancer Research & Treatment, 19 (2020). https://doi.org/10.1177/1533033820916191
[10]. Y. Ji, H. M. Whitney, H. Li, P. Liu, M. L. Giger, X. Zhang, Differences in molecular subtype reference standards impact AI-based breast cancer classification with dynamic contrast-enhanced MRI, Radiology, (2023). https://doi.org/10.1148/radiol.220984
[11]. M. G. Davey, M. S. Davey, M. R. Boland, É. J. Ryan, A. J. Lowery, M. J. Kerin, Radiomic differentiation of breast cancer molecular subtypes using pre-operative breast imaging–a systematic review and meta-analysis, European Journal of Radiology, 144 (2021). https://doi.org/10.1016/j.ejrad.2021.109996
[22]. Y. Zhang, G. Li, W. Bian, Y. Bai, S. He, Y. Liu, H. Liu, J. Liu, Value of genomics-and radiomics-based machine learning models in the identification of breast cancer molecular subtypes: a systematic review and meta-analysis, Annals of Translational Medicine, 10 (24) (2022). https://doi.org/10.21037/atm-22-5986
[23]. M. Caballo, W. B. Sanderink, L. Han, Y. Gao, A. Athanasiou, R. M. Mann, Four-dimensional machine learning radiomics for the pretreatment assessment of breast cancer pathologic complete response to neoadjuvant chemotherapy in dynamic contrast-enhanced MRI, Journal of Magnetic Resonance Imaging, 57 (1) (2023) 97–110. https://doi.org/10.1002/jmri.28273
[24]. T. Zhang, T. Tan, L. Han, L. Appelman, J. Veltman, R. Wessels, K. M. Duvivier, C. Loo, Y. Gao, X. Wang, et al., Predicting breast cancer types on and beyond molecular level in a multi-modal fashion, NPJ Breast Cancer, 9 (1) (2023) 16. https://doi.org/10.1038/s41523-023-00517-2
[25]. O. Lafcı, P. Celepli, P. S. Öztekin, P. N. KoÅŸar, DCE-MRI radiomics analysis in differentiating luminal A and luminal B breast cancer molecular subtypes, Academic Radiology, 30 (1) (2023) 22–29. https://doi.org/10.1016/j.acra.2022.04.004
[26]. T. Huang, B. Fan, Y. Qiu, R. Zhang, X. Wang, C. Wang, H. Lin, T. Yan, W. Dong, Application of DCE-MRI radiomics signature analysis in differentiating molecular subtypes of luminal and non-luminal breast cancer, Frontiers in Medicine, 10 (2023). https://doi.org/10.3389/fmed.2023.1140514
Afterwards, a critical evaluation of the current work against the existing literature must be made, highlighting the novelty of the current work. This aspect should be explained in the Discussion as well.
Response: We have revised the introduction section to highlight the novelty of our work (lines 108-116 in the Introduction section). Since we sought to individually characterize the ER and PR responses, a direct comparison was possible with such studies only, which are fewer in the existing literature. Even fewer existing studies use radiomic features for IDC vs ILC classification. Lines 78-90 in the introduction section evoke some existing studies on individual IHC classifications (references [33, 34] are newly added) as well as IDC vs ILC (references [37, 38] are newly added, references [35, 36] highlight the understudied nature of ILC). Our results against theirs are given in the Discussion subsection ‘4.2. Comparison with existing studies’, lines 394-411.
[33]. H.-J. Yoon, A. Ramanathan, F. Alamudun, G. Tourassi, Deep radiogenomics for predicting clinical phenotypes in invasive breast cancer, in: 14th International Workshop on Breast Imaging (IWBI 2018), Vol. 10718, SPIE, 2018, pp. 391–396. https://doi.org/doi:10.1117/12.2318508
[34]. S. Zhong, F. Wang, Z. Wang, M. Zhou, C. Li, J. Yin, Multiregional radiomic signatures based on functional parametric maps from DCEMRI for preoperative identification of estrogen receptor and progesterone receptor status in breast cancer, Diagnostics, 12 (10) (2022). https://doi.org/10.3390/diagnostics12102558
[35]. J. A. Mouabbi, A. Hassan, B. Lim, G. N. Hortobagyi, D. Tripathy, R. M. Layman, Invasive lobular carcinoma: an understudied emergent subtype of breast cancer, Breast Cancer Research and Treatment, 193 (2) (2022) 253–264. https://doi.org/10.1007/s10549-022-06572-w
[36]. Z. Chen, J. Yang, S. Li, M. Lv, Y. Shen, B. Wang, P. Li, M. Yi, X. Zhao, L. Zhang, et al., Invasive lobular carcinoma of the breast: a special histological type compared with invasive ductal carcinoma, PloS One, 12 (9) (2017). https://doi.org/10.1371/journal.pone.0182397
[37]. K. Holli, A.-L. Lääperi, L. Harrison, T. Luukkaala, T. Toivonen, P. Ryymin, P. Dastidar, S. Soimakallio, H. Eskola, Characterization of breast cancer types by texture analysis of magnetic resonance images, Academic Radiology, 17 (2) (2010) 135–141. https://doi.org/10.1016/j.acra.2009.08.012
[38]. S. Waugh, C. Purdie, L. Jordan, S. Vinnicombe, R. Lerski, P. Martin, A. Thompson, Magnetic resonance imaging texture analysis classification of primary breast cancer, European Radiology, 26 (2016) 322–330. https://doi.org/10.1007/s00330-015-3845-6
Results: the authors mention several times the ‘median value of the mean …’ This is very confusing / erroneous. You either report the median value (which is the middle value of a dataset from smallest to greatest) or the mean value (which represents the average of a dataset). Please clarify these aspects throughout the article.
Response: The mean of three spatial dimensions (in the orientation defined by the acquisition setup) of a tumor is used as a representative of its size. The median is that of these representative sizes. In the revised version we have used the term ‘median size’ to avoid any ambiguity. Our choice of the size representation is mentioned separately. Since the tumors may be quite irregular in shape, we used this indicator of size, rather than the precise 3D volume, for a basic statistical analysis of the homogeneity of tumor sizes. The updated terminology is used in lines 182-187 in subsection ‘2.3. Data preparation and experimental setup’ and lines 277-278 of subsection ‘3.1. Dataset’.
The Discussion must include a section on the limitations of this study – for instance, the large inhomogeneities found among tumour samples, as well as the reduced dataset due to the set inclusion/exclusion criteria are limitations that could impact on the overall analysis and data interpretation.
Response: We have commented on some limitations of our study (subsection ‘4.3. Limitations’, lines 420-435), which include the class unbalances in the datasets, inhomogeneities in the tumors, and lack of a method to accommodate for the variations in the MRI acquisition system (reference [55]).
[55]. S. J. Doran, S. Kumar, M. Orton, J. d’Arcy, F. Kwaks, E. O’Flynn, Z. Ahmed, K. Downey, M. Dowsett, N. Turner, et al., “Real-world” radiomics from multi-vendor mri: an original retrospective study on the prediction of nodal status and disease survival in breast cancer, as an exemplar to promote discussion of the wider issues, Cancer Imaging, 21 (1) (2021) 1–18. https://doi.org/10.1186/s40644-021-00406-6
Specific comments:
Line 43 – replace ‘zone’ with ‘region’
Line 68 – correct ‘intraductal carcinoma’
Line 69 – replace ‘mon-‘ with ‘mono-‘
Line 137 – of the median or of the mean sizes?
Response: We have corrected these elements, which occur in lines 55, 47 (term used earlier), 107, and 183 in the revised manuscript.
Reviewer 2 Report
Journal:JPM (ISSN 2075-4426)
Manuscript ID:jpm-2454561
Type:Article
Title:Characterization of breast tumors from MR images using radiomics and machine learning approaches
Comments:The aim of this study is to explore the predictability of various molecular signatures as well as the histological subtypes intraductal caricnoma (IDC) and intrallobular carcinoma (ILC) in breast tumors via both a mon- and multi-contrast radiomics analysis and, in a limited case, compare it with a CNN-based approach.
The results indicated that the radiomics and CNN-based approaches generally gave comparable results. ER and IDC/ILC classification results were promising. PR and HER2 classifications need further investigation through a larger dataset. The subject of this manuscript is of value, but there are defects need to be modified.
1,Abstract section: Determining histological subtypes, such as intraductal and intralobular carcinomas (IDC 12 and ILC) and immunohistochemical markers, such as estrogen response (ER), progesterone re- 13 sponse (PR), and the HER2 protein status is important in planning breast cancer treatment.
Introduction section: The identification of various immunohistochemical (IHC) markers, like the respon- 31 siveness of the tumors to the HER2 protein, or estrogen (ER) or progesterone (PR) hor- 32 mones, is helpful in determining the course of treatment in breast cancer patients.
Should the author describe ER, PR, and HER2 in the same order throughout the entire text.
2,Figure 2 section: summarizes the categories of the over 300 radiomics features extracted.
It is best to write the accurate number (the over 300 radiomics features).
3,Figure 3 section: The sum of sample sizes in each group, is not equal to the number of cases or lesions described in the text. Please check or explain clearly.
4, 3.1. Dataset section: Although a database of around 500 tumors (from over 300 patients) was available, some 210 factors, such as incomplete histological data, limited the number of tumors available......
from over 300 patients: exact value? Should the author add a Flow chart of the study population with accurate number of cases.
5, 2.5. CNN-based classification section: As a second approach to classifying tumor subtypes, we used a CNN-based classification scheme applied directly to the manually segmented tumors. The full English name of CNN should be provided when CNN first appears.
Author Response
Note: The line numbers referred to in the responses are from the revised manuscript version with the changes tracked. The answers are in blue.
The aim of this study is to explore the predictability of various molecular signatures as well as the histological subtypes intraductal caricnoma (IDC) and intrallobular carcinoma (ILC) in breast tumors via both a mon- and multi-contrast radiomics analysis and, in a limited case, compare it with a CNN-based approach.
The results indicated that the radiomics and CNN-based approaches generally gave comparable results. ER and IDC/ILC classification results were promising. PR and HER2 classifications need further investigation through a larger dataset. The subject of this manuscript is of value, but there are defects need to be modified.
1,Abstract section: Determining histological subtypes, such as intraductal and intralobular carcinomas (IDC 12 and ILC) and immunohistochemical markers, such as estrogen response (ER), progesterone re- 13 sponse (PR), and the HER2 protein status is important in planning breast cancer treatment.
Introduction section: The identification of various immunohistochemical (IHC) markers, like the respon- 31 siveness of the tumors to the HER2 protein, or estrogen (ER) or progesterone (PR) hor- 32 mones, is helpful in determining the course of treatment in breast cancer patients.
Should the author describe ER, PR, and HER2 in the same order throughout the entire text.
Response: We have revised the relevant parts of the manuscript (lines 33-35, Introduction section) to be consistent in the order of these terms.
2,Figure 2 section: summarizes the categories of the over 300 radiomics features extracted.
It is best to write the accurate number (the over 300 radiomics features).
Response: We have indicated the precise number (342) of radiomic features in subsection ‘2.4. Radiomics-based classification’ (lines 199 and 212) in the revised manuscript.
3,Figure 3 section: The sum of sample sizes in each group, is not equal to the number of cases or lesions described in the text. Please check or explain clearly.
Response: The total number of lesions used in this study is 429 (selected after the exclusion criteria). However, each of the ER, PR, HER2, and IDC/ILC statuses is not available for all the tumors. Hence, there is a disparity in the total number of tumors and the number of samples available for a given classification task, shown in Figure 3a. Figure 3b, conversely, is a histogram of the tumor size distribution of all 429 tumors. This explanation has been added at the end of subsection ‘3.1. Dataset ‘(lines 278-282).
4, 3.1. Dataset section: Although a database of around 500 tumors (from over patients) was available, some 210 factors, such as incomplete histological data, limited the number of tumors available......from over 300 patients: exact value? Should the author add a Flow chart of the study population with accurate number of cases.
Response: The initial dataset has 534 tumors from 367 patients. After excluding 78 tumors due to factors such as the unavailability of histological data, and 27 tumors for the stated inclusion criterion for tumor size, we were left with 429 tumors from 323 patients. We have specified this in subsection ‘3.1. Dataset ‘(lines 263-269). Since the exclusion criteria involve just these few steps, we did not include a flow chart.
5, 2.5. CNN-based classification section: As a second approach to classifying tumor subtypes, we used a CNN-based classification scheme applied directly to the manually segmented tumors. The full English name of CNN should be provided when CNN first appears.
Response: The full name, convolutional neural network, for CNN, has been added with the first use of the term (line 16 in the ‘Abstract’ and line 108 in the ‘Introduction’ section).
Round 2
Reviewer 1 Report
The authors have addressed all comments raised by this reviewer. The paper has significantly improved.